# Diagnostic Yield and Safety of CP-EBUS-TBNA and RP-EBUS-TBLB under Moderate Sedation: A Single-Center Retrospective Audit

**DOI:** 10.3390/diagnostics12112576

**Published:** 2022-10-24

**Authors:** Valencia Lim, Reon Yew Zhou Chin, Adrian Kee, Jeffrey Ng, Kay Choong See

**Affiliations:** 1Division of Respiratory and Critical Care Medicine, University Medicine Cluster, National University Health System, 1E Kent Ridge Road, NUHS Tower Block Level 10, Singapore 119228, Singapore; 2Department of Medicine, Yong Loo Lin School of Medicine, National University Singapore, 10 Medicine Drive, Singapore 117597, Singapore

**Keywords:** bronchoscopy, biopsy, diagnosis, endobronchial ultrasound (EBUS), transbronchial lung biopsy (TBLB), mediastinal lymphadenopathy, pulmonary nodule, moderate sedation

## Abstract

Convex probe endobronchial ultrasound transbronchial needle aspirations (CP-EBUS-TBNAs) and radial probe endobronchial ultrasound transbronchial lung biopsies (RP-EBUS-TBLBs) can be performed under moderate sedation or general anesthesia. Moderate sedation is more convenient, however patient discomfort may result in inadequate tissue sampling. General anesthesia ensures better patient cooperation but requires more logistics and also carries sedation risks. We aim to describe the diagnostic yield and safety of CP-EBUS-TBNAs and RP-EBUS-TBLBs when performed under moderate sedation at our center. All patients who underwent CP-EBUS-TBNA and/or RP-EBUS-TBLB under moderate sedation, between January 2015 and May 2017, were reviewed. Primary outcomes were defined in regard to the diagnostic yield and safety profile. A total of 336 CP-EBUS-TBNAs and 190 RP-EBUS-TBLBs were performed between January 2015 and May 2017. The mean sedation doses used were 50 mcg of intravenous fentanyl and 2.5 mg of intravenous midazolam. The diagnostic yield of the CP-EBUS-TBNAs and RP-EBUS-TBLBs were 62.5% and 71.6%, respectively. Complication rates were low with: transient bleeding 11.9%, transient hypoxia 0.5%, and pneumothorax 0.1%. None required escalation of care, post procedure. Performing CP-EBUS-TBNAs and RP-EBUS-TBLBs under moderate sedation is safe and provides good diagnostic yield. These procedures should, therefore, be considered as first-line sampling techniques.

## 1. Introduction

Convex probe endobronchial ultrasound transbronchial needle aspiration (CP-EBUS-TBNA) is widely used in the diagnosis of mediastinal lymphadenopathy [1], mediastinal staging in cancers [2], and molecular testing. It is especially utilized in the context of lung cancer [3]. Radial probe endobronchial ultrasound transbronchial lung biopsy (RP-EBUS-TBLB) is a bronchoscopic modality used to obtain samples of peripheral lung nodules that are determined to be accessible after reviewing computer tomographic scans. It is a modality that provides a good diagnostic yield, while also having a lower risk of pneumothorax complication as compared to percutaneous transthoracic needle aspiration [4]. Both CP-EBUS-TBNAs and RP-EBUS-TBLBs can be combined and conducted sequentially in one procedural session or performed in separate sessions. For both bronchoscopic procedures, options of periprocedural sedation includes general anesthesia and moderate sedation. Periprocedural sedation allows for optimal patient comfort, ensuring that procedures can be carried out smoothly and safely with diagnostic accuracy.

General anesthesia is administered by trained anesthetists, while moderate sedation can be administered by trained endoscopic nurses with instructions from bronchoscopists, or by bronchoscopists themselves. Yarmus et al. reported better patient cooperation when CP-EBUS-TBNAs are performed under general anesthesia, thereby allowing more lymph nodes to be sampled and thus achieving a higher diagnostic yield [5]. A later randomized controlled trial conducted by Casal et al. found that CP-EBUS-TBNAs performed under moderate sedation have comparable diagnostic yields, safety profiles, and patient tolerance to other procedures that are performed under general anesthesia [6]. The use of moderate sedation for the bronchoscopic procedures of CP-EBUS-TBNAs and RP-EBUS-TBLBs have the advantages of cost and logistical arrangement, without the need for artificial airway, anesthetic assistance, additional personnel, drugs, or equipment [7]. Expert centers have reported either favorable outcomes [7,8] or comparable outcomes between endoscopic bronchial ultrasound procedures performed under general anesthesia versus moderate sedation [6,9].

In pursuit of patient-centric practice, our center has routinely performed CP-EBUS-TBNAs with or without RP-EBUS-TBLBs in a single session under bronchoscopist-directed moderate sedation. In this article, we describe our performance of CP-EBUS-TBNAs with or without RP-EBUS-TBLBs in a single session under bronchoscopist-directed moderate sedation, while looking at indications of better diagnostic yields and safety profiles.

## 2. Materials and Methods

A retrospective review of all patients who underwent CP-EBUS-TBNA and RP-EBUS-TBLB between January 2015 and May 2017 at the National University Hospital Singapore (an academic teaching hospital) was performed. The study protocol and waiver of informed consent was approved by the NHG Domain Specific Review Board (DSRB; reference number 2017/00723) given the observational nature of the study. All outpatients and hospitalized patients above the age of 18 years undergoing CP-EBUS-TBNAs and/or RP-EBUS-TBLBs during the study period were included. Indications for both procedures were recorded through the review of case notes. Common indications for CP-EBUS-TBNA included the presence of enlarged mediastinal lymph nodes, suspicions of lung cancer, and the mediastinal staging of lung cancer. Common indications for RP-EBUS-TBLBs included the presence of lung nodules or masses accessible via bronchoscopy. Age, gender, body mass index, comorbidities—such as underlying malignancy, chronic kidney disease, chronic liver disease, and ischemic heart disease—baseline creatinine, coagulation profile, and use of antiplatelets or anticoagulants were also recorded.

All CP-EBUS-TBNA and RP-EBUS-TBLB procedures were performed by an accredited pulmonologist and a specialist trainee. At our center, bronchoscopists undergo 3 years of specialist training and have 3–20 years of postgraduate experience. Procedures are performed with 1–2 bronchoscopist(s), 1 scrub nurse, and 1 float nurse. Parameters such as blood pressure, heart rate, and oxygen saturations are monitored every 5 min. Patients are given topical anesthesia with 1% lignocaine before the induction of moderate sedation with boluses of midazolam, fentanyl, or pethidine that are administered by nurses with instructions from the bronchoscopists. Moderate sedation is administered about 5 min before the start of procedure. At our center, we routinely perform fiberoptic bronchoscopy before CP-EBUS-TBNAs and also followed by RP-EBUS-TBLBs, as required.

CP-EBUS-TBNAs were performed with a real-time ultrasound biopsy bronchoscope (Olympus BF-UC260F, working channel diameter of 2.2 mm). A linear ultrasound transducer (range of 5–12 MHz) was connected to a processor (Olympus EU-ME 1). Transbronchial needle biopsies were performed with a dedicated 22-gauge needle (Olympus NA–201SX-4022) or (Boston Scientific M00558220). Rapid on-site evaluation (ROSE) was performed by a staff pathologist for all EBUS-TBNA procedures. Only 1 type of needle was used per patient.

RP-EBUS-TBLBs were performed using a flexible bronchoscopy (Olympus BFQ290, BF260, working channel diameter of 2 mm), a radial probe (Olympus UM-520-17S diameter 1.8 mm), and single-plane fluoroscopy with an alligator jaw step fenestrated forceps, which were 1.9 mm in diameter (Olympus). A flexible bronchoscopy and a radial probe were both connected to a processor (Olympus CV-260SL or CV-290). A guide sheath was not used for the RP-EBUS-TBLB procedures that were reviewed in this study.

Post procedure, patients were monitored for 3 h and the bronchoscopist would be informed about clinical status of the patient before they are deemed safe for discharge. Patients who underwent RP-EBUS-TBLBs underwent a post procedure chest radiography in order to evaluate for any delayed pneumothorax.

The primary endpoints were the diagnostic yield and safety profiles of CP-EBUS-TBNAs and RP-EBUS-TBLBs. Diagnostic yield was defined by the number of patients who obtained a specific diagnosis from either CP-EBUS-TBNAs or RP-EBUS-TBLBs. Specific diagnosis included primary lung cancer; lung metastasis; caseating granulomatous inflammation indicative of tuberculosis, non-caseating granulomatous inflammation indicative of sarcoidosis; and any other infections. For cases without a specific diagnosis after CP-EBUS-TBNAs and/or RP-EBUS-TBLBs, attempts were made to collect information on follow up or on further investigations that were performed in order to arrive at an eventual diagnosis. Safety profile was evaluated by looking at the incidence of procedure-related complications (such as bleeding requiring the use of ice-cold saline or adrenaline and hypoxia requiring high dependency monitoring or even intubation), sedation-related complications (hypotension, arrhythmia, etc.), and re-admission within a week after the procedure for any delayed complications.

Secondary endpoints were defined as the adequacy of sample and procedure time. Adequacy of sample was defined as a composite endpoint of specific diagnosis, and the presence of either lymphocytes (in the case of CP-EBUS-TBNAs) or lung tissue (in the case of RP-EBUS-TBLBs). Procedure time was defined as the time from administration of sedation till the withdrawal of the bronchoscope.

The primary analysis of the diagnostic yield of CP-EBUS-TBNAs and RP-EBUS-TBLBs were calculated as the percentage of patients for whom the procedures obtained a specific diagnosis. Simple calculations were conducted for characteristics of the sampled lymph nodes and lung masses.

## 3. Results

A total of 421 patients underwent the procedures during the study period between January 2015 and May 2017. Of all the procedures, only one subject was unable to proceed with the procedure despite the maximizing of conscious sedation due to intolerance. A total of 104 patients underwent both CP-EBUS-TBNA and RP-EBUS-TBLB; further, 237 patients underwent CP-EBUS-TBNA only, while 79 patients underwent RP-EBUS-TBLB only. Amongst these patients, a total number of 577 lymph nodes were sampled and 201 lung biopsies were performed. There were 21 bronchoscopists involved, out of whom 6 were specialist trainees. The average number of CP-EBUS-TBNAs performed by each bronchoscopist was 16 (median 2, with an interquartile range of 7). The average number of RP-EBUS-TBLBs performed by each bronchoscopist was 11 (median 3, with interquartile range of 5.75). There were four bronchoscopists specializing in interventional pulmonology, who thus also had the most numbers of procedures.

The mean age of patients was 62 years (Table 1). The mean sedation doses used were 50 mcg of intravenous fentanyl, 2.5 mg of intravenous midazolam, and 50 mg of intramuscular pethidine. The median duration of procedure for each subject was 59 min, with a range of 26 to 135 min.

The top three indications for both procedures of CP-EBUS-TBNA and RP-EBUS-TBLB were mediastinal lymphadenopathy, abnormal findings on lung imaging, and the presence of lung mass (Table 2).

For CP-EBUS-TBNAs, the average number of lymph nodes sampled in each procedure was 1.7 (median 1, range 1–5, and interquartile range 1). Further, the average number of needling per lymph node was 2.38 (median 4, interquartile range 2). The most commonly sampled lymph node stations were in the right lower paratracheal (207 samples), subcarinal (151 samples), left paratracheal (93 samples), right interlobar (42 samples), and right upper paratracheal nodes (24 samples).

For RP-EBUS-TBLBs, 197 lesions were targeted. In six patients, RP-EBUS-TBLBs were performed in two separate lobes, while the rest of the patients had RP- EBUS-TBLBs performed in only one lobe. The average number of biopsy passes was 4.3 (median 5, range 1–15, and interquartile range 3). The average diameter of lung nodules that were sampled was 27.72 mm as measured on the computer tomography scans that were performed prior to procedure (median 37 mm, range 10–125, and interquartile range 29). The most common locations of the target lesion were in the right upper lobe (63 biopsies), right lower lobe (36 biopsies), left upper and lower lobe (30 biopsies each), and right middle lobe (20 biopsies).

The diagnostic yields of CP-EBUS-TBNAs and RP-EBUS-TBLBs were found to be 62.5% and 71.6%, respectively. An adequate sample was obtained in 89.4% of the subjects. The most common diagnosis from both procedures was lung malignancy (Table 3). The most frequent diagnosis for CP-EBUS-TBNAs were malignancy, sarcoidosis, and mycobacterium infection (50.9%, 6.3%, and 4.2%, respectively). The most frequent diagnoses for RP-EBUS-TBLBs were malignancy, organizing pneumonia, and mycobacterium infection (45.8%, 10% and 3.7%, respectively).

In cases where either CP-EBUS-TBNAs or RP-EBUS-TBLBs were undiagnostic, a proportion of patients underwent further diagnostic evaluation—such as transthoracic needle aspirations of the lung nodule, surgical resection, and endoscopic ultrasound fine needle aspiration (EUS-B-FNA), which all obtained the diagnosis.

Complications rates were low at an overall of 12.5%. A total of 50 patients (11.9%) had transient bleeding where hemostasis was achieved by the intrabronchial administration of cold saline, adrenaline, or tranexamic acid. Further, one patient developed pneumothorax after RP-EBUS-TBLB, thereby requiring chest drain insertion. Additionally, two patients (0.5%) had transient hypoxia where oxygenation improved without the need for supplementation post procedure. No subject required escalation of care to the high dependency unit or intensive care unit post procedure. A total of 10 patients (2.4%) were hospitalized for respiratory complaints or fever within 1 week of procedure.

## 4. Discussion

As endoscopic ultrasound procedures become more widely employed around the world, there is an increased interest in the most ideal modality for periprocedural sedation. It is a balance between resource availability, cost consideration, bronchoscopist preference, patients’ comfort, and diagnostic yield [10].

Over the past decade, there have been many studies comparing the diagnostic yield of CP-EBUS-TBNAs that are performed under general anesthesia and moderate sedation. Yarmus et al., in 2013, reported a higher diagnostic yield for CP-EBUS-TBNAs performed under general anesthesia; further they attributed this to the larger number of lymph node sites that were sampled and the greater number needle passes that were performed [5]. However, a randomized, controlled trial conducted by Casal et al., in 2015, comparing general anesthesia to moderate sedation for CP-EBUS-TBNA procedures performed by a single expert bronchoscopist, showed no difference in diagnostic yield; further, it was even reported that a comparatively larger average number of lymph nodes were sampled and a shorter procedure time in the moderate sedation group was found [6]. Studies in recent years report a comparable diagnostic yield of 55–99.5% in procedures performed under moderate sedation and a yield of 77.6–93% in procedures performed under general anesthesia. The incidence of adverse events in procedures performed under both modalities of periprocedural sedation is also similar [11,12,13,14,15,16,17,18].On the other hand, there have not been many studies comparing the diagnostic yield of RP-EBUS-TBNAs performed under general anesthesia versus moderate sedation.

Our study reports a reasonable diagnostic yield, sample adequacy, and low complication rate in the use of bronchoscopist-directed moderate sedation for endobronchial procedures conducted in an academic teaching hospital. To maximize the yield and convenience for patients, CP-EBUS-TBNAs and RP-EBUS-TBLBs were performed in the same session for patients who required both procedures.

We report diagnostic yield and sample adequacy that is consistent with real world data from the AQUIRE registry [7]. Our patient population is an unselected cohort with a mix of benign and malignant conditions. In our center, endobronchial procedures are performed by pulmonologists and trainees of differing levels of training and expertise. Heterogeneity in skills and the involvement of trainees also contributes to our diagnostic yield, sample adequacy, and longer procedural time.

Overall, evidence seems to suggest that diagnostic yield is dependent on multiple factors and the type of sedation may not be a major contributory factor [7,19]. We suggest that procedural technique, operator skills, and combined modalities may overcome potential hinderances caused by the lack of complete cooperation in patients under moderate sedation [8,9]. Expert advocates of the routine use of general anesthesia view time limitation of moderate sedation as a hindrance when performing to complete mediastinal staging in lung cancer [18], whilst expert centers skilled in combined CP-EBUS-TBNA and endoscopic ultrasound fine needle aspiration (EUS-B-FNA) under moderate sedation report additional yield (when EUS-B is added on to EBUS) [20].

Our dual use of CP-EBUS-TBNAs and RP-EBUS-TBLBs in the same session for patients requiring both procedures accounted for a longer procedure duration. Our practice is similar to that of Bailey et al. except for the use of moderate sedation rather than general anesthesia [21]. Our RP-EBUS-TBLB yield for peripheral pulmonary nodules is within the range of published studies [4,22]. We believe that combined modalities incorporating recent advancements and innovations can lead to better diagnostic yields [23].

To increase the diagnostic yield of CP-EBUS-TBNAs under moderate sedation, newer technology can be employed. One example is that of elastography, which uses a color map to identify the areas of lymph nodes to target, which has a reported sensitivity, specificity, positive predictive value, negative predictive value, and diagnostic accuracy rates of 100, 92.3, 94.6, 100, and 96.7%, respectively [24]. In certain cases, the use of a 19 G as compared to a 22 G needle has been reported to increase diagnostic yield from 92%–99%; further, it may be worth considering further patient selection in order to increase diagnostic yield with use of a 19 G needle [25].

To increase the yield of RP-EBUS-TBLBs under moderate sedation, techniques such as virtual navigational bronchoscopy and electromagnetic navigation bronchoscopy have been used. Ishida et al. reported a higher diagnostic yield of 80.4% with virtual-bronchoscopy-assisted EBUS as compared to a diagnostic yield of 67.0% with conventional EBUS- TBLB [26]. A meta-analysis on bronchoscopic techniques by Wang et al., reported a diagnostic yield of 67% with electromagnetic navigation bronchoscopy [27]. Yield of RP-EBUS-TBLB has also been reported to be higher with the use of smaller diameter scopes, the addition of transbronchial needle aspiration of the lesion to biopsy, and the use of thinner and more flexible needles [28]. Oki et al. reported improved diagnostic yield of RP-EBUS-TBLB (74% versus 59%) using an ultrathin bronchoscope of 3 mm diameter (working channel 1.7 mm) versus conventional scope of 4 mm diameter [29]. Thinner bronchoscopes allow the advancement into more peripheral bronchioles, thereby allowing more direct visualization of the peripheral lesion. The addition of transbronchial needle aspirations to biopsy of peripheral lesions have also been reported to increase yield. Arimura et al. compared diagnostic yield between RP-EBUS-TBLBs and TBNAs and found that histological diagnosis of TBNA was comparable to surgical specimens that allowed EGFR mutation testing [30]. However, transbronchial needle aspiration may not be feasible in certain locations, such as upper lobes, as they require a more flexible bronchoscope.

We report a complication rate of 12.5%. All were minor complications and self-limiting. This is comparatively higher than published complication rates [1,6], likely due to the strict reporting of all minor and transient events in our study. None of our patients required escalation of care.

The limitations of our study are found in its retrospective nature and the inability to directly compare general anesthesia vs. moderate sedation. We have no data on patient satisfaction, though studies have shown good satisfaction with endobronchial procedures performed under moderate sedation [8]. Additionally, no formal cost comparison of general anesthesia vs. moderate sedation was performed, though there have been studies which have looked at cost reported cost savings in procedures performed under moderate sedation [13]. Our study contained a heterogeneous group of bronchoscopists, including trainees, which may also have confounded our outcomes. However, this makes our findings more reflective of a real-world practice that is contained in an academic training hospital. Our study also included all patients who underwent CP-EBUS-TBNAs during the study period and did not separately analyze patients who underwent the procedure for the purpose of cancer staging. The yield of CP-EBUS-TBNAs in this group of patients is expected to be higher as the biopsy is conducted in the setting of known malignancies with likely lymph node involvement.

In conclusion, balancing resource utilization, comfort, and convenience while in pursuit of ideals of patient-centric practice, we report reasonable diagnostic yields with a good safety profile in our practice of EBUS procedures under bronchoscopist-directed moderate sedation. Further studies can be conducted in order to look at patient satisfaction with EBUS procedures when performed under moderate sedation.

## Figures and Tables

**Table 1 diagnostics-12-02576-t001:** Patient characteristics (n = 420).

Patient Characteristics	Statistics
Male	279 (66.4%)
Age (years)	62 ± 13
Current/ex smoker	234 (55.7%)
**Existing medical conditions**	
History of malignancy	135 (32.1%)
Chronic kidney disease	32 (7.6%)
Chronic liver disease	6 (1.4%)
Ischemic heart disease	61 (14.5%)
Elevated creatinine (>90 μmol/L)	69 (16.4%)
Thrombocytopenia (<100 × 10^9^/L)	4 (1%)
**Body Mass Index (BMI)**	
Underweight BMI < 18.5	32 (7.6%)
Overweight BMI 25–30	85 (20.2)
Obese BMI > 30	32 (7.6%)
**Coagulopathy**	
PT > 14	
APTT > 35	
**Antiplatelet/Anticoagulation use**	56 (13.3%)
Aspirin	54 (12.9%)
Clopidogrel	
Ticagrelor	69 (16.4%)
Warfarin	22 (5.2%)
Heparin	1 (0.2%)
Rivaroxaban	5 (1.2%)
Ticlopidine	3 (0.7%)
Aspirin + clopidogrel	1 (0.2%)
Aspirin + ticagrelor	1 (0.2%)
Aspirin + dabigatran	6 (1.4%)
	1 (0.2%)
	1 (0.2%)

**Table 2 diagnostics-12-02576-t002:** Indications for either procedure (n = 420 cases).

Indications	Statistics
Mediastinal lymphadenopathy	123 (29.2%)
Abnormal lung imaging (XR/CT)	97 (23.1%)
Lung mass	96 (22.9%)
Cough	22 (5.2%)
Hemoptysis	21 (4.9%)
Non-resolving pneumonia	16 (3.8%)
History of bronchus & lung malignancy	10 (2.4%)
Screening for cancer	7 (1.7%)
Screening for specific respiratory conditions	7 (1.7%)
Screening for infection in immunocompromised hosts	6 (1.4%)
Mediastinal mass for investigation	5 (1.2%)
Suspected mycobacterium infection	5 (1.2%)
Loss of weight	3 (0.7%)
Lung anomaly	1 (0.3%)
Asthma	1 (0.3%)

**Table 3 diagnostics-12-02576-t003:** A: specific diagnosis. B: non-specific diagnosis.

A			
CP-EBUS-TBNA		RP-EBUS-TBLB	
Malignant lung primary	147 (43.8%)	Malignant lung primary	87 (45.8%)
Other malignancies	24 (7.1%)	Organizing pneumonia	19 (10%)
Non-necrotizing granulomatosis inflammation	21 (6.3%)	Mycobacterium	7 (3.7%)
Mycobacterium	14 (4.2%)	Pulmonary metastasis	6 (3.1%)
Bacterial infection	3 (0.9%)	Others: PAP, cyst, cyst, fibrosis	5 (2.6%)
Fungal infection	1 (0.2%)	Non-necrotizing granulomatosis inflammation	4 (2.1%)
		Fungal infection	3 (1.6%)
		Bacterial infection	3 (1.6%)
		ILD	2 (1.1%)
**B**			
**CP-EBUS-TBNA**		**RP-EBUS-TBLB**	
Lymphocytes/non-specific	103 (30.6%)	No specific diagnosis	39 (20.5%)
Reactive changes	23 (6.9%)	Reactive changes	15 (7.9%)

## Data Availability

Not applicable.

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
