# Peer review of "Diagnostic Yield and Safety of CP-EBUS-TBNA and RP-EBUS-TBLB under Moderate Sedation: A Single-Center Retrospective Audit"

_diagnostics, 2022, doi:10.3390/diagnostics12112576_

Round 1

Reviewer 1 Report

This is a retrospective study evaluated the diagnostic yield and safety of CP-EBUS-TBNA and RP-EBUS-TBLB under moderate sedation. The authors demonstrated the good diagnostic yields and low complication rate. My comments were as follows:

Major:

1. Numerous studies on the usefulness and safety of CP-EBUS-TBNA or RP-EBUS-TBLB under moderate conscious sedation have been published. Why did the authors include patients who underwent only CP-EBUS-TBNA and only RP-EBUS-TBLB in this study?

2. Why did the authors include patients who underwent CP-EBUS-TBNA for staging purposes? In patients for staging purposes, the diagnostic yield of CP-EBUS-TBNA largely depends on the prevalence of malignancy. 

Minor:

1. Line 63. Please describe the full form of the abbreviation, “MS.”

2. Line 91, 97. “Olympus BF-UC 260 FW diameter 2.2 mm,” “BF260, with diameter of 2 mm.” It is confusing, “bronchoscope diameter” or “working channel diameter.”

3. Line 91 “A 7.5-MHz linear ultrasound transducer” The range of the ultrasound frequency of BF-UC260FW is from 5MHz to 12MHz.

4. How many patients underwent only CP-EBUS-TBNA? How many underwent only RP-EBUS-TBLB? How many underwent both?

5. Please describe the location of target lesion for CP-EBUS-TBNA (lymph node station) and RP-EBUS-TBLB (lobar location).

6. Please describe the size of the target lesion for CP-EBUS-TBNA.

7. Table 2. Did the authors perform these procedures due to cough, weight loss and asthma?

8. Table 3. Please describe the “specific findings” and “non-specific findings (no specific diagnosis and reactive changes)” separately.

Author Response

Thank you for the review and comments. Kindly see the responses as follow.

Major:

  1. Numerous studies on the usefulness and safety of CP-EBUS-TBNA or RP-EBUS-TBLB under moderate conscious sedation have been published. Why did the authors include patients who underwent only CP-EBUS-TBNA and only RP-EBUS-TBLB in this study?

Response 1: There has not been similar studies published locally in recent years and as one of the academic medical instituition in Singapore, we aim to review the diagnostic yield and safety profile of CP-EBUS-TBNA & RP-EBUS-TBLB performed at our center and how it compares to the yield and safety profiles reported by the published studies. 

  1. Why did the authors include patients who underwent CP-EBUS-TBNA for staging purposes? In patients for staging purposes, the diagnostic yield of CP-EBUS-TBNA largely depends on the prevalence of malignancy. 

Response 2: This study analysed sequential patients who underwent CP-EBUS-TBNA during the study period. The most common indications for procedure as indicated in the bronchoscopy report were: mediastinal lymphadenopathy, abnormal lung imaging and lung mass. Indications of "history of bronchus/lung malignancy" and "screening for cancer" make up <5% of the indications. We acknowledge that the yield of CP-EBUS-TBNA is dependent on the prevalence of cancer, though our study included patients who underwent CP-EBUS-TBNA for all indications and not just for staging. However we admit that the proportion of patients who underwent CP-EBUS-TBNA for staging purposes was not analysed separately from the cohort of patients audited, which may have affected the overall yield; we will address this in the limitations discussion. 

Minor:

  1. Line 63. Please describe the full form of the abbreviation, “MS.”- edited in manuscript
  2. Line 91, 97. “Olympus BF-UC 260 FW diameter 2.2 mm,” “BF260, with diameter of 2 mm.” It is confusing, “bronchoscope diameter” or “working channel diameter.”- edited in manuscript
  3. Line 91 “A 7.5-MHz linear ultrasound transducer” The range of the ultrasound frequency of BF-UC260FW is from 5MHz to 12MHz.- edited in manuscript
  4. How many patients underwent only CP-EBUS-TBNA? How many underwent only RP-EBUS-TBLB? How many underwent both?  - 237 patients underwent CP-EBUS-TBNA; 79 patients underwent only RP-EBUS-TBLTB, 104 patients underwent both procedure . Edited to include in manuscript 
  5. Please describe the location of target lesion for CP-EBUS-TBNA (lymph node station) and RP-EBUS-TBLB (lobar location). For CP-EBUS-TBNA, the most commonly sampled lymph node stations were right lower paratracheal (207 samples), subcarinal (151 samples), left paratracheal (93 samples), right interlobar (42 samples) and right upper pararacheal nodes (24 samples). - edited to include in manuscript. For RP-EBUS-TBLB, the most common lobes of the target lesions were:  right upper lobe (63 biopsies), right lower lobe (36 biopsies), left upper and lower lobe (30 biopsies each) and right middle lobe (20 biopsies). - edited to include in manuscript. 
  6. Please describe the size of the target lesion for CP-EBUS-TBNA. Information on lymph node size was not available 
  7. Table 2. Did the authors perform these procedures due to cough, weight loss and asthma? Indications for procedure was extracted from bronchoscopy report and this is keyed in by performing procedurists. No review of clinical notes was performed to cross check if the indication for procedure keyed in by the procedurist was accurate. 
  8. Table 3. Please describe the “specific findings” and “non-specific findings (no specific diagnosis and reactive changes)” separately. - edited in manuscript 

Reviewer 2 Report

The aim of the work is very important for daily clinical practice in the field of interventional pulmonology.

In my opinion, the study was done very well. The disadvantage of the work is the absence of a comparison group. However, it is clearly stated in the limitations section. On the other hand, a direct comparison of the effectiveness of sedation methods during bronchoscopy would be difficult to accurately assess. There would be some subjectivity.

I have no additional comments. Good job.

Author Response

Thank you for your review and kind comments. 

Round 2

Reviewer 1 Report

Thank you for your revision.